# Analyzing the Impact of Responding to Joint Attention on the User Perception of the Robot in Human-Robot Interaction

**DOI:** 10.3390/biomimetics9120769

**Published:** 2024-12-18

**Authors:** Jesús García-Martínez, Juan José Gamboa-Montero, José Carlos Castillo, Álvaro Castro-González

**Affiliations:** Department of Systems Engineering and Automation, University Carlos III of Madrid, Av. de la Universidad, 30, 28911 Leganes, Spain; jgamboa@ing.uc3m.es (J.J.G.-M.); jocastil@ing.uc3m.es (J.C.C.); acgonzal@ing.uc3m.es (Á.C.-G.)

**Keywords:** Human-Robot Interaction, joint attention, social robots, robot perception, responding joint attention, stimuli, attention

## Abstract

The concept of joint attention holds significant importance in human interaction and is pivotal in establishing rapport, understanding, and effective communication. Within social robotics, enhancing user perception of the robot and promoting a sense of natural interaction with robots becomes a central element. In this sense, emulating human-centric qualities in social robots, such as joint attention, defined as the ability of two or more individuals to focus on a common event simultaneously, can increase their acceptability. This study analyses the impact on user perception of a responsive joint attention system integrated into a social robot within an interactive scenario. The experimental setup involves playing against the robot in the “Odds and Evens” game under two conditions: whether the joint attention system is active or inactive. Additionally, auditory and visual distractors are employed to simulate real-world distractions, aiming to test the system’s ability to capture and follow user attention effectively. To assess the influence of the joint attention system, participants completed the Robotic Social Attributes Scale (RoSAS) after each interaction. The results showed a significant improvement in user perception of the robot’s competence and warmth when the joint attention system was active.

## 1. Introduction

Effective communication and social interaction are built upon sharing focus and understanding each other’s intentions. Individuals constantly coordinate their attention to engage in shared experiences in human interactions and natural environments. Whether following someone’s gaze during a conversation or synchronizing actions in a collaborative task, aligning focus is necessary for establishing understanding, rapport, and cooperation [1]. This capacity, fundamental to human cognition, is equally important in the interactions between humans and artificial agents such as social robots [2].

One of the main elements that support this coordination is joint attention (JA), a complex and essential mechanism that allows individuals to engage with one another by focusing on a shared stimulus, experience or event [3]. JA is a social-cognitive skill that enables individuals to align their focus with others, establishing a shared reference point. This alignment is often initiated or maintained through gaze-following, pointing gestures, or vocal cues. At its core, JA allows individuals to understand and respond to the attentional states of others, creating the foundation for collaboration, social learning, and communication. In human development, JA emerges early in life and plays a central role in communicating and collaborating effectively. Several studies have demonstrated that the deficit of this skill is related to autism spectrum disorder (ASD) [4] and attention deficit hyperactivity disorder (AHDH) [5].

This ability to direct and share attention is not limited to humans [6]. Many animals, especially those living in social groups, rely on JA to navigate their environments and cooperate within their communities. As studied in Emery et al. [7], primates use JA to alert others to potential threats or to coordinate activities such as hunting or foraging. These behaviours in the natural world highlight how JA is a key factor for survival and effective interaction in complex social systems. To create social robots that seamlessly integrate into human society and communicate naturally with humans, we must take inspiration from nature, replicating these fundamental JA mechanisms. According to Mundy et al. [8], JA can be divided into two primary categories: Initiating Joint Attention (IJA) and Responding to Joint Attention (RJA). IJA is an individual’s ability to direct another’s attention to a particular object or event. Communicative cues such as pointing, gesturing, or making verbal sounds often achieve this. For example, a person might point at a distant object and simultaneously look at their interaction partner to ensure they notice it. Alternatively, RJA is the ability to recognise and follow the object or event referred by another person. This implies following the same cues mentioned for IJA. For instance, regarding the previous example, if one person looks at an object and/or gestures toward it, the observer aligns their focus with the indicated direction. Together, IJA and RJA form the building blocks of effective communication and collaborative interaction, enabling shared focus and mutual understanding.

Building on these natural mechanisms, the field of Human-Robot Interaction (HRI) has focused on developing robots that can engage with humans in a socially intuitive and natural way. HRI encompasses various aspects, including communication, social cognition, and the ability of robots to respond to and interpret human behaviours [9]. Many robots struggle with aspects of social cognition, often failing to accurately recognize and respond to human social cues in real-time [10]. By integrating joint attention systems (JAS), robots can better synchronize their focus with their human partners, leading to more natural interactions, as studied in the work of Imai et al. [11].

In particular, this work is centred on RJA. Previous research on RJA has shown that users may interpret a robot’s ability to align attention with theirs as indicating better problem-solving skills or task efficiency. Moreover, robots that respond naturally to social cues can foster smoother interactions, reducing moments of discomfort or misalignment with human norms [12]. Implementing systems with RJA capabilities in social robots presents several challenges. While RJA naturally occurs in humans, replicating it in robots requires the coordination of complex social cues like gaze-following, gestures, and shared intentionality [13]. However, the challenge extends beyond ensuring that robots can technically mimic these behaviours. Until now, few studies have explored how RJA systems shape users’ perception of the robot. This work presents a study aiming to fill this gap by focusing on how users perceive the robot regarding whether RJA is active or not, offering insights into enhancing HRI.

In this study, we integrate a bio-inspired Responsive Joint Attention System (RJAS) into a social robot. This system allows the robot to recognize and respond to user non-verbal cues, such as gaze direction and gestures, in a dynamic gaming scenario that includes auditory and visual distractors the user focuses on. Regarding the bio-inspired parts of the RJAS, our proposed system is modelled after the human vestibular system [14], enabling the robot to shift its focus of attention in a natural, human-like manner and enhancing its responsiveness in social interactions. By replicating human behaviours like gaze shifts and body orientation towards a shared focus of attention (FoA), the system tries to enrich the robot’s capacity to engage in meaningful joint attention with users. Here, the FoA refers to the specific stimulus or area within the environment that an individual prioritises at any given time. For example, a sudden sound or motion may draw immediate attention, overriding less urgent stimuli.

This research’s main contribution lies in evaluating the impact of the RJAS on the user’s perception of the robot during HRI, focusing on metrics that have not been thoroughly analyzed in the context of joint attention previously. Specifically, we examine the user’s perception of the robot’s competence, warmth, and discomfort using the Robotic Social Attributes Scale (RoSAS) [15]. Our experiment compares these dimensions under two conditions: The robot responds to the user’s cues (the RJAS active). The alternative is for the robot to express non-responsive behaviour with random liveliness movements (the RJAS is inactive). The study incorporates auditory and visual distractors to simulate real-world environments, enhancing the realism of the interactions. We collected quantitative data from 91 participants through RoSAS questionnaire and qualitative feedback via an open-response question. This approach provides deeper insights into how users perceive the social capabilities of the robot.

In Section 2, this paper presents the background of our work. It describes the different mechanisms involved in JA and how they can be replicated into a robotic system. Furthermore, we introduce the theoretical background and related implementations necessary for our approach. Finally, previous related works in JA with HRI applications are described. Section 3 details the implementation of the different modules composing the responsive joint attention system. Section 4 describes the experimental setup, including the interactive game, the robotic platform and the distractive stimuli employed to create the gaming scenario. Section 5 details the experimental methods, outlining procedures, tools, and methodologies for developing and evaluating the impact of the system in the robot on user perception. The findings from user experiments and evaluation, encompassing quantitative and qualitative data analysis on system performance and user responses, are summarised in Section 6. Section 7 interprets the results within the context of the study, exploring implications, limitations, and future directions. Finally, Section 8 summarizes key findings, emphasizes the system’s significance in social robots, and suggests avenues for further research.

## 2. Background

To analyze the impact of RJA in HRI, we adopted an interdisciplinary approach, combining insights from various areas of psychology, social cognition, attention and robotics. This allows us to effectively address the human and robotic components, ensuring that the system aligns with natural human behaviours and cognitive processes. The following subsections introduce the theoretical background and explore related studies and implementations. The first subsection addresses studies to determine the most suitable metrics for evaluating the user’s perception of the robot. The second subsection covers various bio-inspired mechanisms of joint attention and their implementations in robotics. Finally, a series of studies implementing JA mechanisms in HRI and their evaluation methods are discussed.

### 2.1. User Perception of the Robot in HRI

The perception that users have of social robots is essential for the success of human-robot interactions. Evaluating attributes related to this concept helps robot developers and designers understand how users interpret and respond to robots, thus facilitating the creation of robotic systems suited to human needs [15]. Well-established theories in social cognition, most notably from research on Stereotype Content Models (SCM) developed by Fiske et al. [16], highlight how humans evaluate others based on attributes such as competence and warmth. These qualities influence and predict how people view and interact with different social groups and are desirable to incorporate into social robots. The SCM was chosen for this study due to its robustness in evaluating social perception dimensions–competence and warmth–that are directly relevant to HRI. Competence reflects a robot’s task efficiency and reliability, while warmth captures its friendliness and approachability. Findings from this framework can affect robotic design by emphasizing how social cues, such as gaze direction or speech patterns, might influence user perceptions. For instance, a robot that maintains appropriate gaze contact could enhance its perceived warmth, while precise task execution might reinforce competence.

**Competence** refers to the perceived ability of an individual or entity to achieve goals efficiently and effectively. In social contexts, competence is often associated with traits like intelligence, skills and reliability, which are necessary for performing tasks successfully [17]. In HRI, competence refers to the user’s perception of the robot’s ability to perform tasks efficiently and reliably. Competence shapes trust and the likelihood of adopting robots for various tasks. For example, in studies like Christoforakos et al. [18], the perception of a robot’s competence has been linked to users’ willingness to rely on them for complex interactions. High competence is associated with increased trust and reduced supervision, making it important for robots involved in decision-making tasks.

**Warmth** captures how friendly, approachable, and trustworthy an individual is perceived. In human social cognition, warmth is often associated with traits like kindness, empathy, and cooperation. According to the SCM, warmth is linked to the perception that groups or individuals seen as less competitive are seen as warmer. This perception influences how people form social bonds and interact, as warm individuals or groups are more likely to be approached and trusted [19]. In HRI, warmth is important in shaping how users perceive social robots. Robots seen as warm are more likely to be viewed as cooperative and engaging, promoting positive emotional connections. Warmth in robots is often communicated through non-verbal behaviours such as facial expressions, gestures, and body language. Robots designed with these characteristics are perceived as more relatable and likeable, which is especially important in roles where social interaction is a key function, such as companionship, or caregiving [20]. Studies in HRI have demonstrated that warmth directly influences the formation of emotional bonds with robots, making it an important metric for promoting long-term human-robot relationships [21,22].

While warmth and competence are well-established metrics in social cognition, the study of Carpinella et al. [15] introduces the discomfort dimension to capture users’ negative emotional reactions. **Discomfort**, as defined by authors, addresses the user’s emotional unease or anxiety when interacting with robots. In some cases, robots perceived as too competent or human-like may provoke discomfort, decreasing willingness to engage with them. The discomfort metric is interesting for identifying the balance between robot design and user comfort, particularly in social robots intended for sustained human interaction.

To evaluate these three dimensions, warmth, competence, and discomfort, Carpinella et al. [15] developed the RoSAS questionnaire. The questionnaire, heavily influenced by the Godspeed questionnaire [23], measures user perceptions of robots across these three dimensions. While the Godspeed items cover various aspects related to the social perception of the robot, including safety and anthropomorphism, the RoSAS focuses on the direct social impact, which facilitates the adaptation and design of robots that interact naturally in social environments. Several recent studies in HRI have examined these attributes using the RoSAS questionnaire, often focusing on different robot types and interaction contexts. For example, in Harris-Watson et al. [24], warmth and competence were evaluated in human-AI teams (HATs), where AI agents were introduced as autonomous teammates. The study explored how human team members’ receptivity to AI teammates depended on these social attributes. The findings revealed that while warmth and competence influenced psychological acceptance and knowledge utilization, competence strongly affected team members’ willingness to integrate the AI’s expertise. Similarly, in Sievers and Russwinkel [25], warmth and competence were measured in an experiment where a social robot spoke either a regional or standard language. The study found that warmth was perceived as significantly higher when the robot spoke in a regional language, demonstrating how language variety can affect user perceptions of a robot’s social attributes. Both studies underscore the importance of measuring these attributes to improve the overall effectiveness of HRI. By integrating these social cognition metrics into our study, we aim to provide a deeper insight into how a social robot with RJA capabilities influences user perception across these dimensions while performing a task, affecting the quality of HRI.

### 2.2. Bio-Inspired Mechanisms of Joint Attention

This section covers various bio-inspired mechanisms of joint attention and their implementations in robotics, serving as the foundation for designing our RJAS. Attention is a cognitive process that allows individuals to selectively focus on specific stimuli from the environment while ignoring less relevant information [26]. This ability to filter and prioritize the FoA is necessary for facing dynamic, real-world situations. For robots, replicating this cognitive process is indispensable to enable full interaction with their surroundings, allowing them to adapt to unexpected situations and successfully engage in JA events. JA unfolds in distinct phases, beginning with establishing a mutual gaze between the participants, as it signals the start of a shared interaction. Once mutual gaze is established, one individual directs the other’s attention towards the FoA by shifting their gaze or pointing towards it. The second participant follows this cue, aligning their attention with the first individual’s FoA. Finally, the participants re-establish eye contact, ensuring that the joint experience is acknowledged and shared, thus closing the loop of JA [27]. These stages are critical for developing social understanding and have been successfully integrated into robotic systems to enrich natural human-robot collaboration [28,29].

Several bio-inspired mechanisms have been considered to replicate these human-like JA behaviours in robots. One of them is **gaze-following**, where the robot tracks and aligns its FoA with the direction of the human’s gaze. This synchronization of attention is essential for seamless joint attention, as shown by Scassellati [30] and Saran et al. [31], where robots equipped with gaze-following capabilities could dynamically adjust their attention based on human cues, creating more natural and responsive interactions. Another JA-related mechanism is **imitation**, which plays a pivotal role in social learning and interaction. Imitation allows robots to replicate human gestures or movements, establishing a shared focus and enriching the social bond between the robot and human [32]. Studies like Ito and Tani [33] have shown that incorporating imitation into JAS significantly improves the robot’s ability to engage in shared tasks, as the robot can mirror actions like pointing or looking, reinforcing the collaborative process. **Turn-taking** is also essential to joint attention, ensuring a balanced and coordinated interaction. By recognizing when to act or respond, robots equipped with turn-taking mechanisms can maintain a smooth flow of conversation or action, as demonstrated by Skantze et al. [34].

In addition to these JA-based mechanisms, a robot should exhibit bio-inspired responsiveness to environmental stimuli and user cues by emulating human behaviours. For example, by replicating how humans direct their gaze toward an object or event using their **vestibular system** [14] or by avoiding repeatedly focusing on previously attended stimuli, known as **inhibition of return (IOR)** [35]. The vestibular system, responsible for coordinating eye and body movements in response to shifts in focus, allows for smooth attention transitions between different focal points through coordinated gaze and body movements. To our knowledge, this mechanism has not yet been implemented into a robotic platform. We hypothesize that robots equipped with vestibular-inspired systems could adjust their posture and gaze fluidly during interactions, mimicking human responses. Additionally, the IOR is a neurocognitive process causing individuals to be less likely to refocus on previously attended stimuli, allowing for more efficient attention shifts to new and relevant stimuli. As evaluated by Sumioka et al. [36], integrating IOR into a robot enables it to maintain a dynamic flow of attention, avoiding distractions from previously attended objects and ensuring continuous engagement with new stimuli. In this work, we propose integrating some of the previously described bio-inspired mechanisms –gaze-following, imitation, turn-taking, IOR, and vestibular movement– in our RJAS to improve the robot’s ability to engage in JA with humans. By leveraging these biological principles, the robot can prioritize stimuli, manage attention shifts, and maintain a natural interaction flow, providing a more intuitive and engaging HRI experience.

### 2.3. Joint Attention in Human-Robot Interaction

In this section, we review studies that have implemented JA mechanisms in HRI, focusing on how they evaluate the impact of their systems across various dimensions. These studies highlight the role of JA in improving interaction quality, using different approaches to measure both technical performance and user experience. By exploring this body of work, we aim to identify gaps in the existing research to guide our study on JA’s impact on HRI. These gaps directly motivate our study, which aims to address this need through a bio-inspired system and rigorous evaluation using standardised metrics. In a work by Imai et al. [11], researchers explored how a speech generation system could incorporate JA to enhance human-robot interaction. The study integrated eye contact and physical attention gestures such as pointing to guide participants in shared tasks. The system uses attention coordinates to represent both the robot’s and participant’s focus of attention, helping develop JA more naturally. This approach was validated through a psychological experiment, showing that eye contact played a significant role in establishing JA and improving communication between humans and robots. Huang and Thomaz [37] investigated how a robot’s ability to RJA affects performance during a labelling task. Participants interacted with the Simon robot [38], which responded to referential gestures or remained focused on the user. The task involved associating colours with objects, with participants teaching the robot the correct associations through speech and pointing gestures. The results showed significantly better performance in the RJA group, fewer errors and redundant labels, and a better understanding of the robot’s internal state. Participants in the RJA group also perceived the robot as more socially interactive, supported by questionnaire results and video analysis of participants’ behaviour.

In the study of Skantze et al. [34], the authors researched the role of turn-taking and gaze-following in an HRI task. The experiment involved a face-to-face setting where the robot provided route instructions to participants for drawing on a map. The robot’s gaze was used to manage JA, focusing on landmarks on the map to facilitate the participants’ understanding. The study explored different conditions: one where the robot exhibited random gaze behaviours, another where the robot was hidden, and a consistent face-to-face interaction with focused gaze cues. The authors found that JA via gaze-following significantly improved task performance and participant satisfaction, particularly in disambiguating landmarks. The robot’s gaze also strongly influenced participants’ turn-taking behaviour, as it helped them decide when to act or give feedback during the task. In a work by Pereira et al. [39], the authors focused on how RJA affects user perception during a problem-solving task. They implemented an autonomous system that uses multimodal inputs such as gaze direction, speech, and actions to simulate JA behaviours. Their findings showed that robots employing RJA increased feelings of social presence among participants, especially when the robot’s gaze behaviours aligned with user actions. They measured social presence using standardized social presence questionnaires (SPI). Similarly, Mishra and Skantze [40] proposed a planning-based gaze control system designed to automate gaze behaviours in social robots. Their system incorporated mechanisms such as turn-taking and gaze-following, dynamically adjusting gaze priorities based on the state of the interaction. Evaluated through a card-sorting game scenario, users rated the planned system significantly higher regarding intimacy regulation than the baseline. Woertman [41] investigated the effect of JA on trust. Participants interacted with two robots, one engaging in JA and another in disjoint attention. Trust levels were measured right after the interaction and again after one week. While no significant effect on trust was found due to JA, initial trust scores were shown to predict trust levels strongly in the second session. The study highlighted the potential influence of likability over JA in shaping trust, emphasizing the need for further research to refine the evaluation process. This study used a custom questionnaire to evaluate trust and engagement metrics, focusing primarily on gaze-following mechanisms implemented in the robot’s JAS.

These studies reveal challenges in implementing joint attention mechanisms in HRI. One of the main limitations is scalability, as many systems are tested in environments that do not account for the variability of real-world settings. In this sense, we aim to replicate a scenario where the user is exposed to stimuli that may arise during their interaction with the robot, encouraging them to shift their attention. This approach allows for a more effective evaluation of whether the attention mechanisms implemented in the robot are functioning as intended. Another observed pattern is the difficulty in ensuring robustness across different robotic platforms, as the hardware capabilities of robots significantly influence how the researcher is able to integrate JA mechanisms on the platform. Existing works often rely on custom or ad hoc robotic platforms explicitly designed for their experiments. This limits the general applicability of their findings to other systems.

Table 1 summarizes the aforementioned studies related to HRI incorporating JA capabilities, primarily focusing on RJA. These studies evaluate various aspects of HRI, including metrics like social presence, trust, task performance, and engagement, using different methods such as video records, questionnaires, and custom tools. Our system is differentiated by combining JA-based mechanisms, such as gaze-following and imitation, and by considering bio-inspired mechanisms, such as the vestibular system and the IOR phenomenon. By incorporating these mechanisms, our system replicates human-like attention behaviours and improves the interaction’s naturalness. Our study analyses different aspects of social cognition- including competence, warmth, and discomfort- and evaluates them through the RoSAS standardised questionnaire and user feedback. One of the key findings in this review is the absence of a standardized baseline for evaluating JA in HRI. There is a clear diversity in the metrics measured and the tools employed across these studies. Our contribution fills part of this gap by leveraging bio-inspired models to enhance responsiveness and realism in robot behaviours and by employing diverse metrics, such as competence, warmth, and discomfort, to offer a broader evaluation of JA’s role in HRI.

## 3. Responsive Joint Attention System

The RJAS aims to expand the robot’s capability to naturally respond to user cues in real time, specifically targeting head and body gestures. This system allows the robot to track user attention and dynamically adjusts its behaviour to engage in RJA. The primary purpose of the RJAS is to enhance the naturalness of HRI by replicating bio-inspired mechanisms based on psychological and biological concepts of social cognition and emulate human attention behaviours.

The proposed system’s architecture (shown in Figure 1) relies on several **user feature detectors** to interpret user-related cues. These detectors include the following (see Figure 2): (i) Face Detector: This detects the user’s face, providing 3D coordinates from the robot’s RGB-D camera and the bounding box in the image, enabling the robot to calculate the user’s position in real-world coordinates (see Figure 2a). (ii) Head Pose Detector: This module estimates the orientation of the user’s head using Euler angles (yaw, pitch, and roll) to determine where the user is looking. This allows the robot to mimic human gaze-following behaviour (see Figure 2b). These two detectors belong to the OpenVino Toolkit (https://www.intel.com/content/www/us/en/developer/tools/openvino-toolkit/overview.html (accessed on 17 November 2024)). Then, the (iii) Body Pose Detector (https://github.com/google-coral/project-posenet (accessed on 17 November 2024)) estimates the landmarks that form the human body, considering the arms to calculate the direction in which the user points (see Figure 2c). Finally, the (iv) Hand Detector (https://ai.google.dev/edge/mediapipe/solutions/vision/hand_landmarker (accessed on 17 November 2024)) tracks the user’s hand position, detecting whether the fingers are extended or retracted, which is particularly relevant for the chosen game (see Figure 2d). These detectors allow the RJAS to calculate the user’s focus, enabling the robot to engage in RJA behaviours.

The RJAS is a two-staged system; the first stage involves a module that computes the visual attention inspired by state-of-the-art concepts [42], obtaining the user’s FoA based on their cues. This module is called **Visual Attention Module (VAM)**. Afterwards, the **Bio-inspired Behavioural Module (BBM)** computes the angles for each robot’s joints to reorient the robot towards the FoA. The VAM processes sensory information to assign intensity values to each detected cue through the **stimuli intensity manager** module, which calculates the FoA based on these weighted cues. Intensity values, ranging from 0 to 1, are dynamically adjusted according to the interaction context. For instance, primary JA signals such as gaze direction and arm gestures are assigned an intensity of 1 when the user turns their head or points with their arm, indicating priority. The user’s face is assigned a baseline intensity 0.5, providing a constant reference point. In specific activities, such as games requiring hand signals, the intensity for hand cues is elevated to 0.8, ensuring their prominence over the user’s face during the different game phases. These values were set empirically inspired by how humans prioritize different types of attention-related stimuli [43].

The system stores multiple measurements of each stimulus to increase reliability, ensuring consistency before making a decision. If a stimulus variates its value during several frames, meaning it is inconsistent throughout time (i.e., due to the detector malfunctioning), the system discards it. Additionally, the system includes a logic filter that discards contradictory cues; for example, if the user points to the right while moving their head to the left, both stimuli are disregarded, and the robot focuses on the user’s face or the user’s hand accordingly. The VAM also integrates an IOR-inspired mechanism. When the robot attends to a particular stimulus, its intensity is set to zero, allowing it to revert its focus to the user’s face, which retains a constant intensity value and acts as a stable reference. This mechanism supports natural gaze-shifting behaviour as the robot alternates between new stimuli and the user’s face, emulating the gaze-shifting behaviours seen in humans.

When a user cue is detected, the VAM calculates the final FoA and sends this information to the BBM, which then adjusts the robot’s movements to align with the user’s attention. The BBM handles both the robot’s movements and verbal responses. First, the **motion** controller implements a bio-inspired imitation mechanism, where the robot replicates the user’s head movements during gaze-shifting to maintain shared attention on specific objects, enhancing the robot’s ability to follow user gestures naturally. In addition, the BBM’s motion controller shown in Figure 3 integrates a simulation of the vestibular system, ensuring smooth coordination between the robot’s eyes, head, and body. When the robot shifts its gaze, it moves its eyes (Figure 3a), turns its head (Figure 3b), and finally rotates its torso (Figure 3c), just as humans do. The robot only moves its head and torso if the detected FoA is outside its current field of view, mimicking the vestibular reflex in humans. Additionally, the eyes perform a counter-movement to stabilize the gaze during head movements, further enhancing the naturalness of the robot’s behaviour.

The system also incorporates an **attention verbal expression manager** module, which allows the robot to select predefined statements to inform the user when it has detected a distraction that lasts more than 1.5 seconds. This feature enhances the interaction by making the user aware that the robot is responsive to their actions, reinforcing the sense of engagement and making the interaction more dynamic. The robot’s verbal expressions are randomly selected from a predefined set, avoiding repetition. Some examples of these verbal responses include phrases like, “It seems you’ve seen something interesting”, or “It looks like something has caught your attention”. This strategy helps the user to realise that the robot is actively monitoring and responding to their behaviour, thereby maintaining a fluid and natural interaction.

## 4. Experimental Setup

This section presents the elements of our experimental setup, which were designed to evaluate the integration of bio-inspired mechanisms into a social robot. Our study aims to assess how these mechanisms affect user perception and interaction quality during a dynamic, task-based game, specifically in the context of JA. The setup integrates various components, such as a social robot platform that integrates the RJAS, a set of distractive stimuli and a game for interactive tasks.

### 4.1. Social Robot Mini

Mini [44] is a desktop social robot designed and built by the Social Robotics Lab at Carlos III University of Madrid, and it serves as the platform for the experiment conducted in this study. Initially created to assist and entertain older adults, Mini features a friendly appearance as depicted in Figure 4, designed to encourage face-to-face interaction with users, promoting HRI in various environments, such as private homes and residences. While initially developed with a focus on physical and cognitive stimulation for the elderly to improve their well-being, Mini’s applications have expanded to tackling issues such as undesired loneliness and isolation or reducing the digital gap in elders.

In terms of hardware, Mini has five degrees of freedom in its head, neck, arms, and torso, allowing it to perform non-verbal gestures. It features an RGB-D depth camera (RealSense D435i (https://www.intelrealsense.com/depth-camera-d435i/) (accessed on 17 November 2024)) in its body to capture 3D information about the environment and the user. Capacitive touch sensors on the torso, arms, and head detect user interactions, while LEDs in the cheeks, heart, and mouth simulate heartbeats and affective expressions. Mini’s eyes consist of two uOLED screens that display natural expressions. Audio is managed through microphones in the head and torso to capture ambient sound and the user’s voice, along with a speaker for verbal communication. To improve AI processing, Mini integrates Google Coral TPU (https://coral.ai/products/accelerator (accessed on 17 November 2024)) and Intel Movidius Neural Compute Stick 2 (https://www.intel.la/content/www/xl/es/products/sku/140109/intel-neural-compute-stick-2/specifications.htmlr (accessed on 17 November 2024)) accelerators, reducing CPU load and supporting local execution of AI models. An auxiliary tablet provides multimedia displays, allowing for dynamic user interaction.

### 4.2. Distractive Stimuli

In our study, the distractive stimuli were carefully designed to test the effectiveness of the RJAS during the game. The purpose of these distractions was to mimic real-world environmental stimuli that might naturally divert a participant’s attention, thereby providing a dynamic and realistic setting to evaluate how the robot detects and responds to such shifts in attention.

The distraction setup comprises two main components: auditory and visual stimuli. According to the literature, auditory distractions tend to capture human attention more effectively than visual stimuli [45]. Therefore, more auditory stimuli were used in the experiment to challenge the participant’s ability to maintain focus. The auditory distractions were delivered through speakers positioned discreetly behind an opaque screen, ensuring the sound source was unclear and thus more likely to capture attention. These sounds included noises such as drum beats, bells, and breaking glass, each introduced at specific points in the game to simulate random environmental noises.

In addition to the auditory stimuli, visual distractions were presented on a monitor to the side of the table, within the participant’s peripheral vision. Research suggests that moving visual stimuli attract more attention than static images, so we opted for animated GIFs over still images [46]. These visual cues included animations such as a moving vehicle or a cartoon chase scene designed to capture the participant’s visual attention. These stimuli were strategically timed to appear when the user’s focus was expected to be on the game, testing their ability to stay engaged despite competing visual information.

The distractors simulate real-world events where social robots might need to recognize and react to attention shifts caused by external events. The hierarchy of stimuli, where auditory distractions generally have a higher impact on human attention than visual stimuli and where moving visuals are more engaging than static ones, justifies our selection of stimuli types [43]. Through this setup, we aimed to evaluate how effectively the robot’s RJAS could reorient the user’s attention back to the task at hand.

Table 2 summarizes the distractive stimuli used in the study, specifying their source, type, duration, and description. This setup provides a controlled environment to evaluate the robot’s joint attention capabilities. By incorporating these distractive elements, we expect to assess how well the RJAS handles distractions and re-engages users, simulating conditions typical of everyday human interactions.

### 4.3. Odds and Evens Game

The main goal of the experimental setup is to create an immersive and realistic environment for HRI to evaluate the impact of RJA on the user’s perception of the robot. For that, we decided to integrate the robot with the RJAS and distractive stimuli into an experimental scenario where the user plays with the robot in the “Odds and Evens” game. The “Odds and Evens” game is a classic chance-based game where two participants must choose either “odds” or “evens”. Following a synchronised countdown, each player extends several fingers, and the sum of the fingers determines the outcome. If the total is an even number, the player who chose “evens” wins; if the total is odd, the “odds” player wins.

Figure 5 illustrates the interaction flow, starting with the robot **presenting** itself and **explaining the game rules** if needed. A **calibration phase** follows, where the robot ensures that the user is familiar with the display of numbers using their fingers. The game then proceeds with the user or robot selecting whether they will play as “odd” or “even” during the match and **reminds** the user of the basics of interacting with the robot to avoid false positives with the finger detection. Then, Mini guides the interaction, actively managing the **countdown** and informing the user of the game progress in real-time.

After the countdown, Mini displays its choice on the tablet, and the user has to **display their hands**. If the user does not show their fingers in time or the robot does not detect them accurately, the robot alerts the user and resets that round. Otherwise, the robot **checks who the winner** of this round is. This sequence repeats for five rounds until the match concludes with a winner. After each match, the robot checks if the user wishes to continue playing, concluding with a farewell when the **game ends**.

Auditory and visual distractive stimuli are introduced at different game stages to simulate a real-world scenario where the robot must respond to user distractions. These stimuli are carefully distributed throughout the interaction—**reminder**, **countdown** and **check winner** phases– to avoid overstimulating the user and ensure that the primary focus remains on the game itself.

## 5. User Study

This section describes the methodology employed to investigate the impact of the RJAS integrated into the social robot, Mini, on the robot’s perception from the user standpoint. We developed a between-subject design user study to test two experimental conditions related to the absence or presence of the RJAS. The resulting conditions are as follows:*RJAS Active*: Participants play the “Odds and Evens” game with the robot while the joint attention system is active. Visual and auditory distractions try to capture the user’s attention while the user is playing the game. If the user gets distracted, the robot will turn towards the user and express verbally that it noticed the distraction.*RJAS Inactive*: A similar setup, but without the robot reacting to the user’s distractions.

### 5.1. Participants

A total of 91 volunteers participated in the experiment, comprising 47 men and 44 women. The age distribution was as follows: 48 participants aged 18–24, 27 aged 25–34, 5 aged 35–44, 10 aged 45–54 and 1 aged 55–64. Participants were randomly assigned to one of the two conditions to ensure a balanced distribution across the experimental setups. Demographic data, including age, gender, and current mood, were collected at the beginning of the session to control for potential confounding variables.

### 5.2. Procedure

The experiment was conducted in a controlled environment with the robot Mini placed on a table alongside auxiliary devices. The robot’s tablet was positioned in front to avoid being interpreted as a distraction by the robot, allowing accurate detection while the user viewed game instructions. Distractions were introduced through two speakers hidden behind an opaque screen to obscure sound sources and a monitor on the right side of the table displaying visual stimuli. This setup, illustrated in Figure 6, simulated real-world distractions with sounds from one speaker at a time and a series of GIFs designed to divert the participant’s attention.

The experiment began with the experimenter delivering a comprehensive briefing to each participant, informing them about the study’s objectives and the procedures they would follow. Following this introduction, the volunteer was seated and asked to complete a data protection form. This document required them to provide their personal information, contact details, and a unique identifier (assigned by the experimenter) that would serve as their pseudonym throughout the study. Once the participant had reviewed and signed the consent form, they were invited to engage in an initial interaction with Mini. At this point, the experimenter exited the room, allowing the participant to interact with Mini independently. The next phase of the experiment involved Mini and the participant playing the “Odds and Evens” game together. This implied two matches playing best-of-five rounds against the robot. Then, after finishing the whole interaction, the session concluded with participants being asked to complete a post-experiment questionnaire.

A session involving two matches and all the distractors for one of the volunteers was recorded on video as a demonstration (Video of a user with the RJAS Active condition: https://youtu.be/piXg4xh4_mI (accessed on 17 November 2024)).

### 5.3. Post-Experimental Questionnaire

We used the RoSAS for the post-experimental questionnaire, which measures warmth, competence, and discomfort through 18 items. Each dimension is represented through nine descriptive terms that help participants reflect their impressions accurately. **Warmth** includes items that assess how friendly and approachable a robot is perceived, which can influence trust and openness in interactions. **Competence** focuses on evaluating the robot’s perceived effectiveness and reliability, impacting users’ confidence in the robot’s ability to perform tasks. **Discomfort**, on the other hand, comprises items that gauge negative feelings, such as apprehension or unease, which could hinder sustained engagement. Responses were collected using a 5-point Likert scale.

At the end of the questionnaire, we added a control question asking whether the participants noticed anything that distracted their attention from the robot and, if so, what it was. In addition, we included an open and non-compulsory `Comments’ question, where the participants could give their opinion about the study and the elements involved, such as the distractive stimuli, the system’s performance or the robot’s behaviour.

### 5.4. Hypotheses

The independent variable in this study is whether or not the RJAS is active in the robot. The dependent variables are the dimensions of the user’s perception of the robot measured using the RoSAS, specifically Competence, Warmth and Discomfort. By manipulating the activation of the RJAS, the objective is to evaluate how the RJA mechanisms influence participants’ perceptions and interactions with the robot. Based on this premise, we formulate the following hypotheses:**H0:** There will be a difference in the **participants’ perception of the robot** depending on whether the RJAS is active or not.**H1:** Participants in the active RJAS condition will rate the robot as having higher **competence** than those interacting with a robot without RJAS. The idea motivating this hypothesis is that when the robot appears to understand and synchronize its actions with the participant’s FoA, it might improve the interaction’s fluency and give the impression of greater problem-solving and task-oriented abilities [37].**H2:** Participants interacting with a robot with an activated RJAS will report higher **warmth** from the interaction with the robot than those interacting with a robot without RJAS. We expect that using an RJAS might increase the robot’s perceived warmth by encouraging more natural, human-like social interactions [39].**H3:** Participants interacting with a robot with an activated RJAS will report lower levels of **discomfort** than those interacting with a robot without RJAS. We hypothesise that the active RJAS could avoid awkward interactions by aligning the robot’s behaviour with human social norms, therefore lowering perceived discomfort [47].

## 6. Results

This section presents the statistical analyses carried out after the experiments. The tool used to obtain the results was IBM SPSS Statistics software (version 26). We used the G*Power software (version 3.1.9.7) to perform a post hoc power analysis to assess the power achieved with our sample. We set an alpha significance level of α=0.05 for the analyses.

### 6.1. Quantitative Results

Given the nature of the experiments, which involved two distinct conditions and a between-subject design, we employed independent sample t-tests for normally distributed data. When this assumption was not satisfied, we utilized the Mann-Whitney test instead. Our analysis focused on comparing the mean scores across various dimensions of participants’ perception of the robot between the two conditions: those interacting with a robot with an active RJAS and those engaging with a robot with the RJAS turned off.

The analysis begins by examining the **competence** dimension across the entire population involved in the experiment. To ensure the robustness of the analysis, we tried to identify potential outliers. The interquartile range for the variable was used, but no cases were detected. Therefore, the dataset included 91 samples, 46 from the active RJAS condition and 45 from the inactive RJAS condition. First of all, we proceeded to conduct Shapiro-Wilk tests to check normality per dimension. Concerning competence, this dimension follows a normal distribution (p=0.335). After testing the normality, we conducted independent samples t-tests to compare the competence reported between participants interacting with a robot with the active RJAS and those interacting with a robot without it. The results showed that there was no significant difference in competence between the interaction with the active RJAS (M=4.246, SD=0.533) and the inactive system (M=4.059, SD=0.579) conditions. The effect size, calculated using Cohen’s *d*, was d=0.337, indicating a small to medium effect size. A post-hoc power analysis conducted using G*Power, with an alpha of 0.05, demonstrated that the t-test for competence with the current sample size (N=91) had a power of 0.479.

After analysing competence, we proceeded with **warmth**. Again, we proceeded to detect and remove outliers from the dataset. Analyzing the interquartile range for each dependent variable, we identified and removed three cases considered outliers according to SPSS criteria. The final dataset comprised 88 samples, 44 for each condition. The Shapiro-Wilk tests revealed that this dimension followed a normal distribution (p=0.123). The *t*-test results indicated significant differences between both conditions of the experiment (active system, M=3.504, SD=0.618, and inactive system, M=3.133, SD=0.729); t(86)=2.576, p=0.012. For this dimension, the effect size was d=0.549, suggesting a medium effect size. The post-hoc analysis revealed that the t-test for warmth with the current sample size (N=88) achieved a power of 0.819 for detecting a medium effect size.

Finally, we identified and removed another three outliers for the **discomfort** dimension. The final dataset comprised 88 samples, 43 from the active RJAS condition and 45 from the inactive RJAS condition. The Shapiro-Wilk tests revealed that the discomfort dimension did not follow a normal distribution (p<0.001). We then used a Mann-Whitney test to examine the discomfort. In this case, the results did not reveal significance between the active RJAS (M=1.198, SD=0.210) and the inactive RJAS (M=1.341, SD=0.397) conditions. The effect size was obtained through Rosenthal’s *r* for this dimension. The result was r=0.134, suggesting a small effect size. The post-hoc analysis revealed that the Mann-Whitney for this dimension with the current sample size (N=88) achieved a power of 0.150 for detecting a small effect size. All the results for the three dimensions are shown in Figure 7.

In addition to the general results, we conducted a more detailed analysis considering participant gender. In this sense, we believe this balance between male and female participants provides an interesting opportunity to explore potential differences in user’s perception of the robot based on gender. The dataset for male participants included 47 samples, with 27 in the active RJAS condition and 20 in the inactive RJAS condition. Similarly, the dataset for female participants comprised 44 samples, with 19 in the active RJAS condition and 25 in the inactive RJAS condition.

The outlier analysis did not detect any cases regarding the **competence** dimension across **male participants**. The Shapiro-Wilk tests to check normality per dimension revealed that this dimension followed a normal distribution (p=0.818). The independent sample *t*-tests reported that there was a significant difference in competence between the interaction with the active RJAS (M=4.309, SD=0.450) and the inactive system (M=3.908, SD=0.681) conditions; t(45)=2.427, p=0.019. The effect size was d=0.716, indicating a medium to large effect size. The post-hoc power analysis revealed that the results had a power of 0.772.

Regarding **warmth** in male participants, the outlier analysis allowed removing three cases considered as such. The resulting dataset comprised 44 samples, 25 from the active condition and 19 corresponding to the inactive RJAS condition. The Shapiro-Wilk tests revealed that this dimension followed a normal distribution (p=0.980), and the t-test results indicated significant differences between both conditions of the experiment (active system, M=3.640, SD=0.524, and inactive system, M=3.061, SD=0.821); t(42)=2.848, p=0.007. The effect size for this dimension in male participants was d=0.867, and the power was 0.876.

We removed one case in the male population for the **discomfort** dimension. The final dataset comprised 46 samples, 26 from the active RJAS condition and 20 from the inactive RJAS condition. The Shapiro-Wilk tests revealed that the discomfort dimension did not follow a normal distribution (p=0.003). We then used a Mann-Whitney test to examine the discomfort. In this case, the results did not reveal significance between the active RJAS (M=1.218, SD=0.230) and the inactive RJAS (M=1.392, SD=0.460) conditions. For this dimension, the effect size was r=0.138. The post-hoc analysis revealed that the Mann-Whitney for this dimension achieved a power of 0.115 for detecting a small effect size. Figure 8 shows the results for the three dimensions regarding the male participants.

Regarding the **female population**, the outlier analysis did detect one case regarding the **competence** dimension. The resulting dataset comprised 43 samples, 19 from the active condition and 24 corresponding to the inactive RJAS condition. The Shapiro-Wilk tests revealed that this dimension followed a normal distribution (p=0.481). The independent samples t-tests reported that there was not a significant difference in competence between the interaction with the active RJAS (M=4.158, SD=0.635) and the inactive system (M=4.222, SD=0.419) conditions. The effect size was d=−0.123, and the post-hoc power analysis revealed that the results had a power of 0.105.

Concerning **warmth** in female participants, the outlier analysis identified one case. The resulting dataset comprised 43 samples, 19 from the active condition and 24 corresponding to the inactive RJAS condition. The Shapiro-Wilk tests displayed that this dimension followed a normal distribution (p=0.148), and the *t*-test results indicated there were no significant differences between both conditions of the experiment (active system, M=3.281, SD=0.774, and inactive system, M=3.256, SD=0.575). The effect size for this dimension in female participants was d=0.035, and the power was 0.063.

For the **discomfort** dimension, we removed two outliers. The final dataset comprised 42 samples, 17 from the active RJAS condition and 25 from the inactive RJAS condition. The Shapiro-Wilk tests showed that the discomfort dimension once again did not follow a normal distribution (p=0.001). The results from the Mann-Whitney test did not reveal significance between the active RJAS (M=1.167, SD=0.177) and the inactive RJAS (M=1.300, SD=0.344) conditions. For this dimension, the effect size was r=0.146, indicating a small effect size. The post-hoc analysis showed a power of 0.115 for detecting a small effect size. Figure 9 shows the results for the three dimensions regarding the female participants. For a detailed summary of the analysis results, Table 3 contains the averages, standard deviations, *p*-values, effect sizes, and power for each condition across different dimensions and participant types.

Finally, we also addressed the responses regarding the control question related to the participants realising the presence of the distractors during their interaction with the robot. Of the 91 participants in the experiment, 76 reported noticing the distractors, and 41 of them were from the RJAS active condition. This accounts for 83.51% of the overall participants and 89.13% of those in the RJAS active condition. On the other hand, 15 participants (3 in the RJAS active condition) either did not notice the distractors or were unsure about their presence.

### 6.2. Qualitative Results

Of the 91 participants, 69 used the `Comments’ section of the post-experiment questionnaire to provide feedback. Two independent raters analysed the comments, and they were classified into five categories: Positive, negative, constructive, error reports and neutral observations. The main themes of the comments were focused on (i) the interaction with the robot, (ii) the distractions introduced during the game, (iii) the timing and flow of the experiment and (iv) detection issues regarding the hand detector.

Seventeen participants highlighted the experiment as an engaging and enjoyable experience, seven being part of the active RJAS condition. One participant remarked: “It was a very good experience. Very fun and entertaining” (ID = 7, RJAS Active), reflecting the generally positive reception of the interaction. Others appreciated how the robot contributed to the game’s dynamic nature, such as “The experiment is very well designed. As time passes, you truly forget that you’re interacting with a machine and feel like you’re simply playing” (ID = 3, RJAS Active). However, three volunteers reported the robot being too abrupt or even harsh when distracted: “When you shift your attention, instead of seeming like it’s following you, it feels like it’s scolding you” (ID = 51, RJAS Active).

Another sixteen participants commented on how the robot handled distractions and attention. One of them commented “I liked it quite a lot. If you didn’t pay attention to the robot, it would quickly get your attention to bring you back to the game” (ID = 11, RJAS Active). Another user praised the robot in this aspect: “The robot noticed when I looked away or lost attention, which was very well done” (ID = 34, RJAS Active). Despite this, some volunteers found the robot too sensitive, detecting distractions even when looking forward (ID = 15, RJAS Active) or looking directly at the robot’s eyes (ID = 48, RJAS Active). Another participant mentioned that even slight head movements were misinterpreted as distractions (ID = 81, RJAS Active).

The timing of the countdown to reveal fingers was a common concern across both conditions. One participant in the RJAS Not Active condition mentioned that the countdown between “1, 2, 3” and “GO” felt uneven, making them unsure about when to reveal their fingers and leading to concerns about the robot potentially cheating (ID = 19, RJAS Not Active). Similarly, in the RJAS Active condition, another participant suggested that the countdown should clearly indicate when to reveal the fingers to avoid confusion (ID = 49, RJAS Active).

Finger detection issues were mentioned in both conditions by 18 participants. One noted that the robot sometimes failed to correctly count the number of fingers they showed, such as counting an extra finger (ID = 79, RJAS Active). There were also reports of the robot misreading two-handed gestures (ID = 40, RJAS Not Active) and making occasional counting errors during games (ID = 25, RJAS Not Active).

## 7. Discussion

As the results indicate, the activation of the RJAS significantly affects participants’ perceptions across different dimensions. While not all dimensions revealed the same significance level, the overall findings help reveal the role of the RJAS in enhancing HRI, supporting hypothesis **H0**.

Regarding the Competence dimension, the results did not show improvement when the RJAS was active for the whole population. However, the analysis by gender showed a significant improvement for this dimension (p=0.019). For this reason, we could say that hypothesis **H1** was partially supported. In addition, the post-hoc power analysis indicated moderate statistical power (0.772), providing confidence in the robustness and replicability of these findings. Concerning the results from the complete population and the women groups, the power obtained was 0.479 and 0.105, respectively, indicating a potential risk of Type II error. To address this, we suggest an analysis with a larger sample size and a more balanced age group distribution. We believe the RJAS’s capacity to align the robot’s gaze with the participant’s FoA and engage in mutual gaze exchanges enhanced the robot’s perceived competence by creating a sense of dynamic responsiveness. This could also be interpreted as the robot demonstrating attentiveness, which participants may have interpreted as tracking their intentions in real-time. This behaviour heightened the robot’s perceived problem-solving ability and suggested understanding the participant’s focus, indicating situational awareness. This is supported by previous studies that emphasised the importance of JA in improving perceptions of competence in robots [37].

Similarly, significant differences were found in the Warmth dimension across all the participants (p=0.012), where participants interacting with the active RJAS reported the robot to be warmer and more engaging, proving hypothesis **H2**. This would be consistent with previous findings describing how RJA enhances users’ perception of the robot’s friendliness and approachability by promoting more natural and human-like interactions [39]. Furthermore, the post-hoc power analysis indicated a power of 0.819 for our study. The gender-based analysis showed that male participants perceived the robot as significantly warmer (p=0.012), whereas female participants perceived it as warmer without reaching statistical significance. As it happened with the competence dimension, the results did not show enough statistical power for the female group to discard a Type II error. In addition to the results for competence, the findings for warmth suggest a potential interaction between the participant’s gender and their sensitivity to perceiving the robot’s competence or warmth, depending on whether the system is active. This possibility is supported by studies that not only evaluate these aspects but also examine the interaction between the participant’s gender and the gender they attribute to the robot [48]. Although it would be interesting to conduct deeper research into this aspect, we consider it falls outside the scope of this work.

With respect to the discomfort dimension, the Mann-Whitney tests did not reveal a noticeable reduction in this dimension’s levels between the active and inactive RJAS conditions (p>0.05), which means hypothesis **H3** was not supported in any case, regardless of the gender of the participants. This suggests that while the RJAS contributes to perceptions of competence and warmth, it may not impact the discomfort during the interaction. This fact could be attributed to other factors not directly addressed by the system, such as the participants’ pre-existing attitudes toward robots, the complexity of the task or the social robot’s appeal due to its appearance [47]. Also, the statistical power of the results for this dimension revealed that future studies with a larger sample size could potentially show significant differences between conditions.

Regarding the control question, we observed that most participants (83.51%) noticed the distractors, adding validity to the experimental design and emphasizing its effectiveness in simulating a realistic environment. This result indicates that the experiment successfully engaged participants’ attention and provided insights under conditions that mimic real-life scenarios.

The qualitative feedback gathered in the questionnaires reinforces some of the quantitative findings while revealing areas for improvement. Many participants found the interaction engaging, especially in the active RJAS condition, aligning with the higher Warmth scores. However, some noted that the robot’s responses to distractions felt abrupt, which may explain why Discomfort did not decrease as expected. This suggests that while the RJAS enhances engagement, it may need adjustments to avoid negative emotional responses.

A limitation of our study is that it tests RJA without adding IJA or a combination of both. In this sense, we plan to examine how the IJA alone and IJA and RJA together might interact and contribute to the overall perception of robot competence, warmth, and discomfort. In addition, the study was conducted using one robotic platform. While the RJAS system itself was the primary focus, the physical appearance and design of the robot could have influenced participants’ perceptions of Competence, Warmth, and Discomfort. The robot’s size, shape, or expressiveness might have affected how participants evaluated the robot’s abilities and social interaction quality.

In addition to this, implementing JA mechanisms such as gaze-following and imitation in real-world systems presents technical and conceptual challenges. From a technical standpoint, achieving real-time operation requires the execution of an entire robotic architecture alongside the attention system. Our robot operates solely on CPU resources without internal GPU acceleration, thus requiring highly optimised vision models to sustain an acceptable frame rate during real-time interaction. Furthermore, vision-based head orientation estimations inherently involve some error due to the model itself, which propagates through the system and can affect the robot’s attentional responses. On a conceptual level, replicating JA mechanisms in the robot does not guarantee that users perceive them as intended. Evaluating whether these mechanisms create the feeling of JA remains subjective and is challenging to measure. While questionnaires like RoSAS assess user perceptions of competence and warmth, tools like the Social Presence Inventory (SPI) could provide additional insights into whether users truly perceive shared focus from the robot.

Another limitation involves the control question introduced in the questionnaire. Initially, we considered using the question as a criterion for considering participants in the active condition as outliers since they might not have experienced the RJAS. We finally decided that the current phrasing, which asks about the perception of distractors, was not precise enough for this purpose. In our opinion, the question should be revised from `Did you notice anything that distracted your attention? If so, what?’ to `Did you perceive that the robot responded to your shifts in attention during the interaction?’ or something similar. This change would enable us to detect better whether the user had the opportunity to experience the system truly. Lastly, regarding the limitations of this work, the study’s short-term nature restricts our understanding of the long-term effects of RJAS. Future research should include longitudinal studies with repeated interactions to assess how RJAS influences user perceptions and behaviour over time.

Regarding more possible research directions, subsequent studies could expand on the findings of this work by applying the RJAS to different types of games and interactive scenarios. This way, we can explore how each game’s specific dynamics, interaction style and cognitive demands may influence user perception and the system’s effectiveness. Additionally, while this study focused on a single robotic platform, evaluating the system across various social robots with different levels of expressiveness and attributes could help isolate the system’s impact from the robot’s inherent features. Another interesting research path could involve integrating machine learning techniques to enhance the RJAS, allowing the robot to adapt and learn from user behaviour to align with human attention patterns according to different visual stimuli.

## 8. Conclusions

This study investigated the impact of an RJAS on user perception during HRI, focusing on three key dimensions: Competence, Warmth, and Discomfort. A total of 91 participants interacted with a robot under two conditions, “RJAS Active” and “RJAS Not Active”, within a game scenario that simulated real-world distractions. Participants’ perceptions were measured using the RoSAS to evaluate how the RJAS influenced their experience.

The results demonstrated that the RJAS significantly enhances the perception of the robot’s Warmth, and partially the Competence. Male participants interacting with the robot with the active RJAS reported higher Competence scores, indicating that the system improved the robot’s ability to respond to user actions, which enhanced their sense of task-related performance. The Warmth dimension showed significant improvements for the whole population, with participants perceiving the robot as more engaging and friendly when the RJAS was active. However, no significant differences were found in the Discomfort dimension, suggesting that while the RJAS positively influences warmth and competence, it may not substantially reduce discomfort during interactions. This could be due to the need for further adjustments in the robot’s responses to distractions or additional design improvements.

The study also collected qualitative feedback, which aligned with the quantitative findings. Many participants found the interaction enjoyable and immersive, particularly in the active RJAS condition, further supporting the improvements in warmth. However, some noted that the robot’s reactions to distractions could be abrupt, which may explain why discomfort levels remained unchanged. These insights highlight the need to refine the robot’s responsiveness to create a more seamless and natural interaction.

While this study provides strong evidence of the positive effects of RJAS on user perception, limitations should be addressed in future work. Firstly, the study tested RJA in isolation, without considering IJA or a combination of both. Future research should explore the interplay between IJA and RJA, as both mechanisms likely contribute to different aspects of user perception. Moreover, according to the results concerning statistical power, future studies with a larger sample size could help reveal statistically significant differences in the female population for the three dimensions. Finally, conducting longitudinal studies with repeated interactions would help understand the long-term effects of RJAS on user engagement and perception over time.

We hope the findings presented in this article highlight the design implication that integrating RJAS into social robots may improve their perception as warmer and more competent. In this sense, we believe attributes like these might be necessary for encouraging positive interactions and acceptance in diverse social settings, consequently contributing to improving HRI.

During the preparation of this work, the authors used GPT-4o, from OpenAI, and Grammarly, from Grammarly Inc., to improve the work’s readability and language. After using these tools, the authors reviewed and edited the content as needed and take full responsibility for the publication’s content.

## Figures and Tables

**Figure 1 biomimetics-09-00769-f001:**
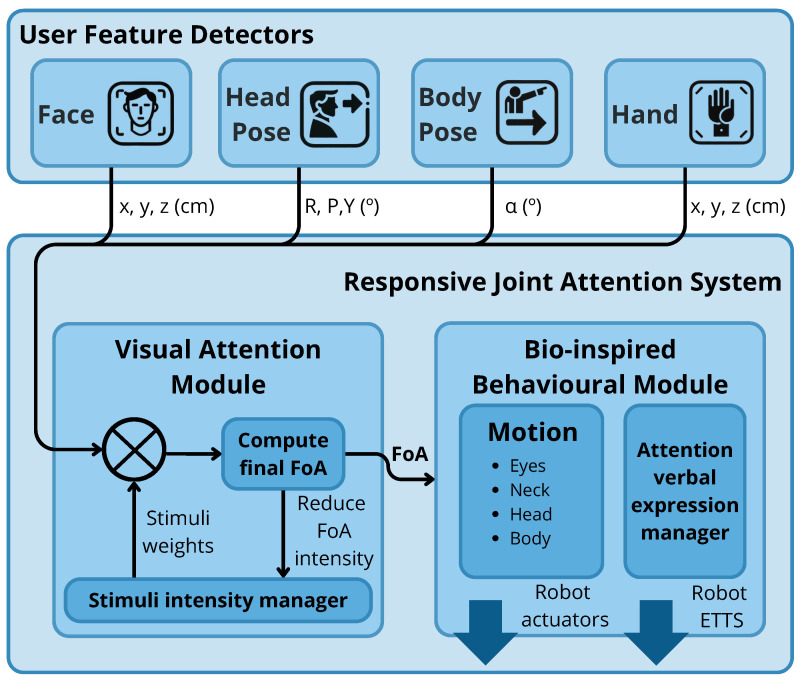
Overview of the RJAS. The system considers detector stimuli input and applies weights to compute the FoA, which the motion controller uses to reorient the robot and perform verbal expressions.

**Figure 2 biomimetics-09-00769-f002:**
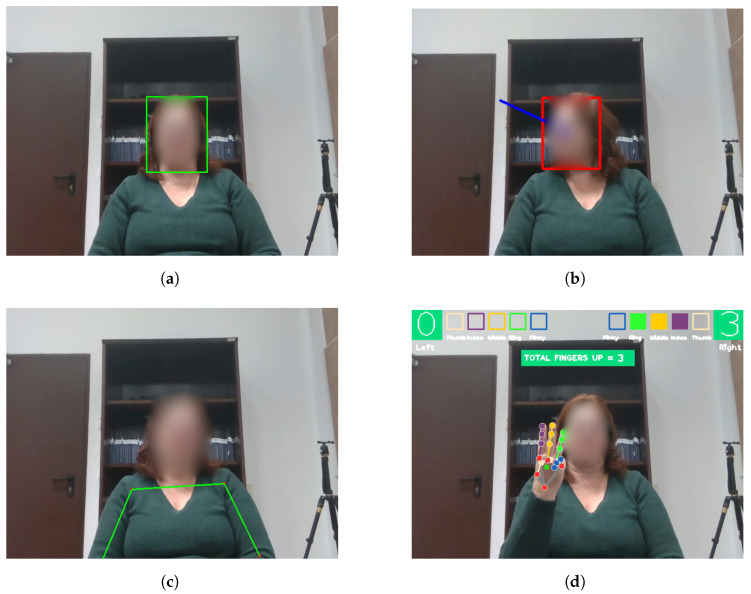
Different frames of one of the users were captured by the detectors through the robot’s camera during the experiment. (**a**) Face Detector. The user’s face is highlighted in a green bounding box. (**b**) Head Pose Detector. The user’s gaze direction is represented with a blue arrow. (**c**) Body Pose Detector: The user’s body landmarks are highlighted with green arrows and red dots. (**d**) Hand Detector: The hand’s finger landmarks and the current finger status are displayed in the image.

**Figure 3 biomimetics-09-00769-f003:**
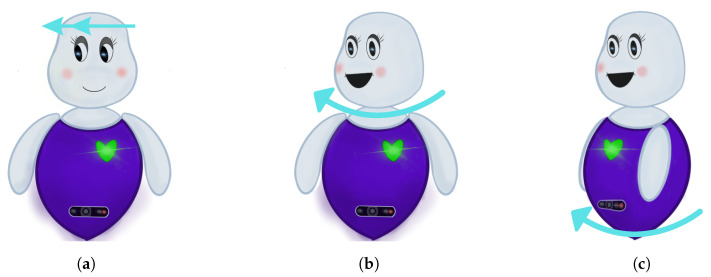
BBM controller phases imitating the vestibular system mechanism. (**a**) The robot starts by moving the eyes toward the FoA if it’s within its field of view. (**b**) It turns its head toward the direction of the FoA while doing a counter-movement with the eyes. (**c**) Finally, the robot reorients the body towards the FoA.

**Figure 4 biomimetics-09-00769-f004:**
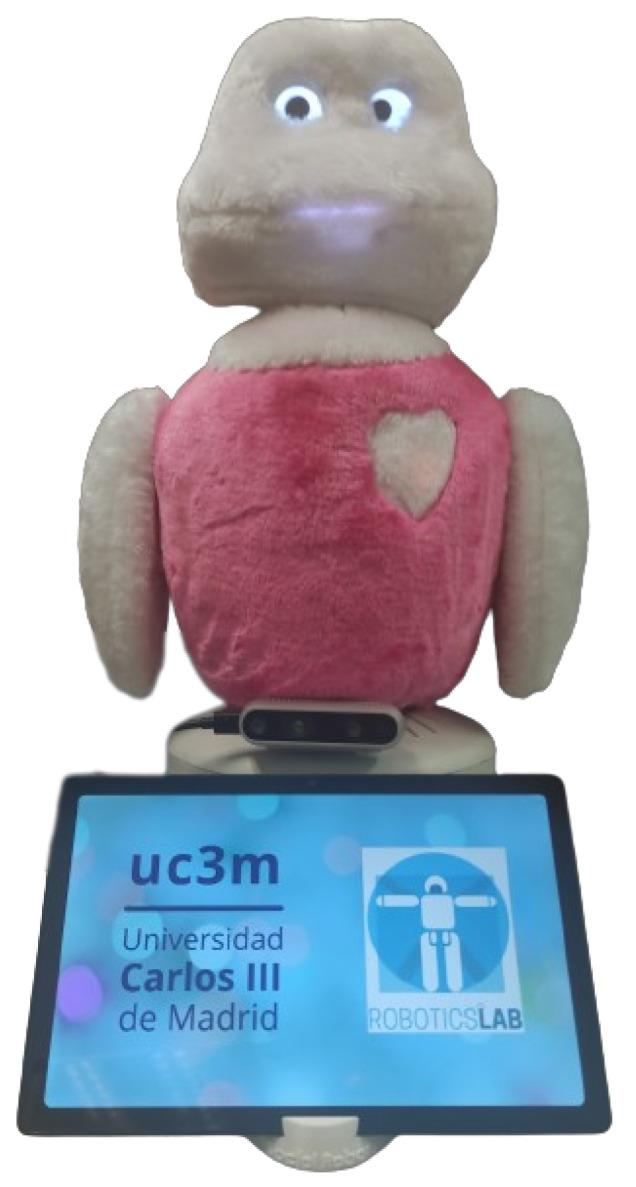
The social robot Mini.

**Figure 5 biomimetics-09-00769-f005:**
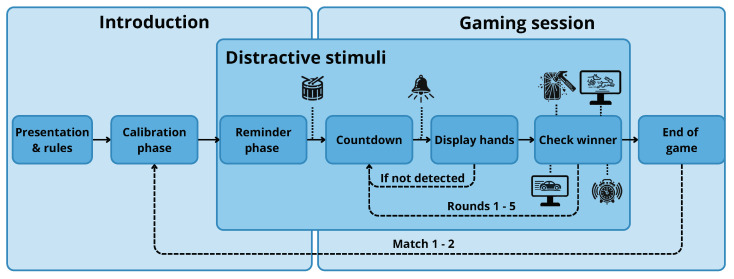
Workflow of the gaming experience. The blue boxes represent the common game stages in the scenario. Distinctive icons represent the different stimuli (audio and visual).

**Figure 6 biomimetics-09-00769-f006:**
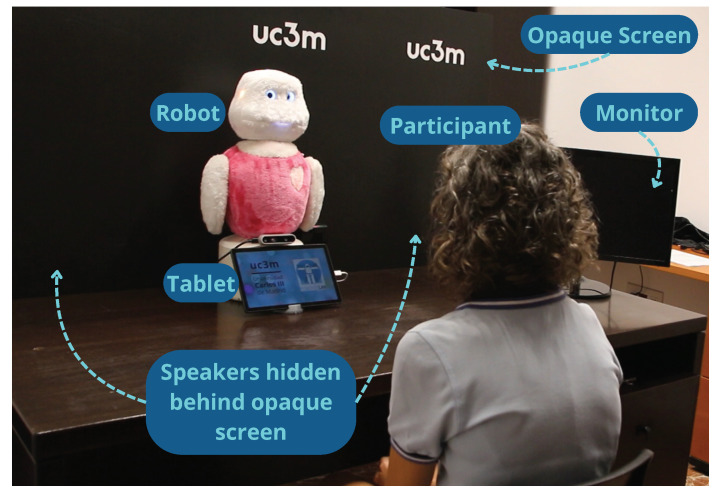
Study setup.

**Figure 7 biomimetics-09-00769-f007:**
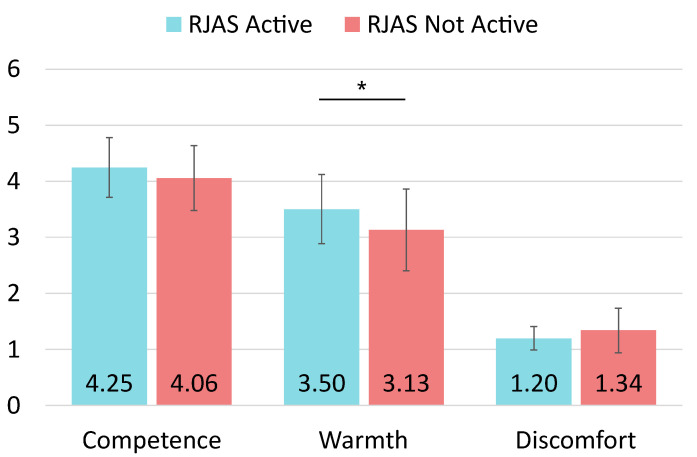
Bar charts with the average scores and SDs for each of the three dimensions of the user’s perception of the robot measured by the RoSAS questionnaire. Significance levels are indicated: * for p<0.05.

**Figure 8 biomimetics-09-00769-f008:**
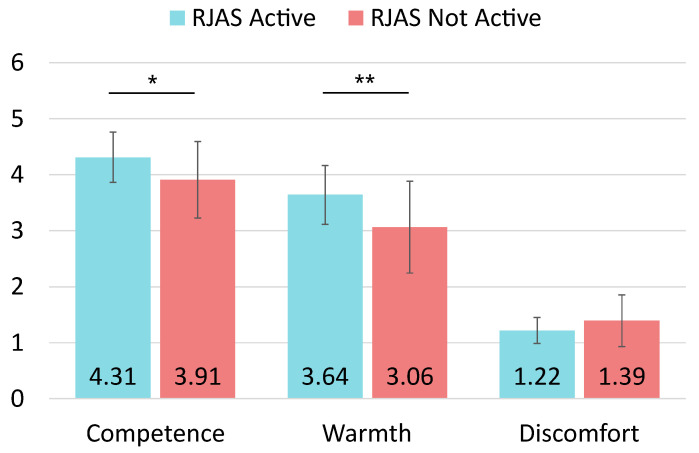
Average scores and SDs for male participants’ perception of the robot. Significance levels are indicated: * for p<0.05 and ** for p<0.01.

**Figure 9 biomimetics-09-00769-f009:**
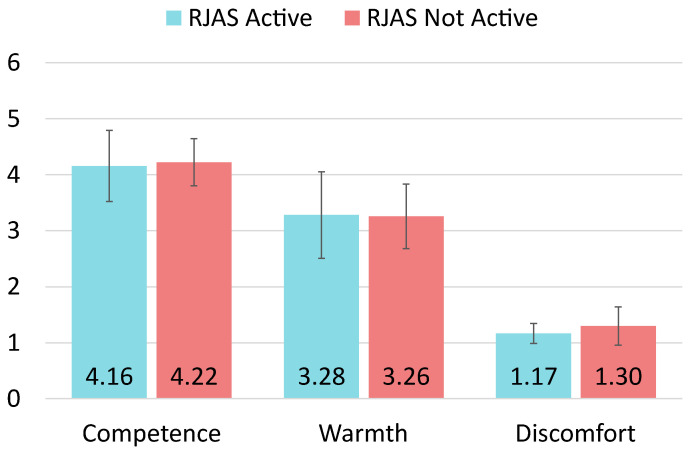
Average scores and SDs for female participants’ perception of the robot.

**Table 1 biomimetics-09-00769-t001:** Summary of Joint Attention Studies in Human-Robot Interaction.

Reference	Users	Joint Attention Mechanisms	Metrics	Data Source
Imai et al. [11]	20	Eye Contact, Pointing gestures	Achievement of JA	Video
Huang and Thomaz [37]	20	Pointing gestures, Gaze-following	Task performance, Engagement	Video, Custom questionnaire
Skantze et al. [34]	24	Turn-taking, Gaze-following	Task performance, Drawing activity	Quantitative data, Custom questionnaire
Pereira et al. [39]	22	Gaze-following	Social presence	SPI questionnaire
Mishra and Skantze [40]	26	Turn-taking, Gaze-following	Awareness, Human likeness, Intimacy	Custom questionnaire
Woertman [41]	40	Gaze-following	Trust	Custom questionnaire
**Ours**	**91**	**Gaze-following, Imitation, Turn-taking**	**Competence, Warmth, Discomfort**	**RoSAS, User feedback**

**Table 2 biomimetics-09-00769-t002:** Summary of Distractive Stimuli Used in the Study.

Source	Duration	Type	Description
Left speaker	2 s	Auditory	Sound of a drum
Right speaker	5 s	Auditory	Sound of bells
Monitor	Until round finishes	Visual	Animation of a moving vehicle
Left speaker	1 s (twice)	Auditory	Sound of breaking glass
Monitor	Until round finishes	Visual	A chase between a coyote and a rabbit
Right speaker	5 s	Auditory	Sound of an alarm clock

**Table 3 biomimetics-09-00769-t003:** Summary of results across different populations, dimensions of the RoSAS questionnaire, and experimental conditions. Statistically significant values are shown in bold.

Population	Dimension	Condition	N	Average	SD	*p*	Power
General	Competence	Active	46	4.246	0.533	>0.05	0.479
		Inactive	45	4.059	0.579		
	Warmth	Active	44	3.504	0.618	**0.012**	**0.819**
		Inactive	44	3.133	0.729		
	Discomfort	Active	43	1.198	0.210	>0.05	0.150
		Inactive	45	1.341	0.397		
Men	Competence	Active	27	4.309	0.450	**0.019**	**0.772**
		Inactive	20	3.908	0.681		
	Warmth	Active	25	3.640	0.524	**0.007**	**0.876**
		Inactive	19	3.061	0.821		
	Discomfort	Active	26	1.218	0.230	>0.05	0.115
		Inactive	20	1.392	0.460		
Women	Competence	Active	19	4.158	0.635	>0.05	0.105
		Inactive	24	4.222	0.419		
	Warmth	Active	19	3.281	0.774	>0.05	0.063
		Inactive	24	3.256	0.575		
	Discomfort	Active	17	1.167	0.177	>0.05	0.115
		Inactive	25	1.300	0.344		

## Data Availability

For access to the data, please contact any of the authors using the contact information provided in the paper.

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
