# Peer review of "Analyzing the Impact of Responding to Joint Attention on the User Perception of the Robot in Human-Robot Interaction"

_biomimetics, 2024, doi:10.3390/biomimetics9120769_

Round 1
Reviewer 1 Report
Comments and Suggestions for Authors
This study analyses the impact on user perception of a responsive joint attention system integrated into a social robot within an interactive scenario. Auditory and visual distractors are employed to simulate real-world distractions, aiming to test the system’s ability to capture and follow user attention effectively.
1. The problem solved in the manuscript is relevant. The authors have described very well in subsection 7.1 the future research based on the results of the experiment. Indeed, when one reads the article, there are similar suggestions and comments that one should have still tried this way or that. Therefore, limiting myself to this study, I omit a number of remarks, as a more detailed experiment is needed, which is what the authors are planning. There are some minor comments and suggestions though.
2. I would suggest to the authors for future research, to compare the results between the two different in character games. In addition, it is worth analyzing attributes and qualities (variables) in a complex (system analysis) with a search for cause-and-effect relationships between variables.
3. It is also “begs” to use artificial intelligence in the training of the robot for customization.
4. Considering the speed with which methods and technologies in robotics are developing, only a quarter of the sources in the reference list are new (for the last 5 years). More up-to-date research should be included in the review.
5. References to Figures in the text should be made before the figures themselves. Similarly with Tables.
6. Normally one subsection (7.1) is not numbered.
Author Response
We have updated the manuscript content after carefully addressing all the reviewers' comments. We are uploading an updated manuscript where the changes have been highlighted using blue for you and olive for the other reviewer.
Comments 1: The problem solved in the manuscript is relevant. The authors have described very well in subsection 7.1 the future research based on the results of the experiment. Indeed, when one reads the article, there are similar suggestions and comments that one should have still tried this way or that. Therefore, limiting myself to this study, I omit a number of remarks, as a more detailed experiment is needed, which is what the authors are planning. There are some minor comments and suggestions though.
Response 1: We thank the reviewer for their positive feedback and for recognising the relevance of our work. The joint attention system we developed provides a new framework to systematically analyse key gaps in joint attention and human-robot interaction for social robots. In this sense, we are committed to conducting further experiments to explore and address these open questions.
Comments 2: I would suggest to the authors for future research, to compare the results between the two different in character games. In addition, it is worth analyzing attributes and qualities (variables) in a complex (system analysis) with a search for cause-and-effect relationships between variables.
Response 2: We agree with the reviewer in this regard. Exploring the impact of the system across games with different dynamics is an important direction, as aspects like interaction style and cognitive demands could influence the results. Similarly, while the RoSAS questionnaire already evaluates the relationships between its metrics, we acknowledge the value of analysing the system's impact on various robotic platforms with different physical attributes, such as expressiveness or appearance, to better isolate the system's effects from the overall impact of the robotic platform.
Action 2: We have included the following ideas summarised in the discussion (Section 7):
"Regarding more possible research directions, the following studies could expand on the findings of this work by applying the RJAS to different types of games and interactive scenarios. This way, we can explore how each game's specific dynamics, interaction style and cognitive demands may influence user perception and the system's effectiveness. Additionally, while this study focused on a single robotic platform, evaluating the system across various social robots with different levels of expressiveness and attributes could help isolate the system's impact from the robot's inherent features."
Comments 3: It is also “begs” to use artificial intelligence in the training of the robot for customization.
Response 3: We greatly appreciate the reviewer’s suggestion on this matter. Currently, in parallel, we are working on an update to the JAS to further align the robot's visual attention with how human visual attention operates. We are employing eye-tracking technology to detect where the user looks in response to various visual stimuli. This information will be used to train a learning model. However, it is important to note that this improvement is still under development and, therefore, out of this publication's scope.
Action 3: We have included a mention of the use of artificial intelligence to update the attention system in future iterations to Section 7: “Another interesting research path could involve integrating machine learning techniques to enhance the RJAS, allowing the robot to adapt and learn from user behaviour to align with human attention patterns according to different visual stimuli.”
Comments 4: Considering the speed with which methods and technologies in robotics are developing, only a quarter of the sources in the reference list are new (for the last 5 years). More up-to-date research should be included in the review.
Response 4: We thank the reviewer for this observation. We agree that keeping the state of the art updated is essential. We have to clarify that the use of older references in this study is primarily due to the foundational nature of the concepts being addressed. Despite this, following the reviewer's remark, we have reviewed the state of the art and identified additional and more recent relevant studies that align with our contributions. These have been included in the manuscript.
Action 4: Reference [3] in Section 1, pointing to the work by Mundy et al (2007), has been updated to the work by Moll (2024), having a more recent definition of Joint Attention. Reference [28] from Section 2, change from the work by Admoni et al. (2017) to Stephenson et al. (2021) since it also constitutes an example of a robotic system that integrates eye contact. Finally, we added the work of Mishra et al. to the list of the most relevant JA experiments in Section 2. This involved updating Table 1 with another research item and adding the following text to the manuscript. “Similarly, Mishra & Scantze [40] proposed a planning-based gaze control system designed to automate gaze behaviours in social robots. Their system incorporated mechanisms such as turn-taking and gaze-following, dynamically adjusting gaze priorities based on the state of the interaction. Evaluated through a card-sorting game scenario, users rated the planned system significantly higher regarding intimacy regulation than the baseline.”
Comments 5: References to Figures in the text should be made before the figures themselves. Similarly with Tables.
Response 5: We thank the reviewer for their observation. We have made the necessary adjustments to ensure that references to figures and tables were placed before them in the text, or at least, in cases where this was not possible, that these elements are placed in the main text as near as possible to the first time they are cited.
Action 5: We have adjusted the placement of references to figures and tables to ensure they appear before the elements and are placed in the main text. In cases where multiple figures appear stacked because of this rule, we put the reference after the figure or table causing the issue, always ensuring the closeness between both elements.
Comments 6: Normally one subsection (7.1) is not numbered.
Response 6: We appreciate the reviewer’s observation. The decision to number subsection 7.1 was made to better structure and organize the extensive discussion, particularly regarding limitations and future work. However, we acknowledge the reviewer's point and have removed the subsection title, merging its content into the main discussion section.
Action 6: We have removed the title of subsection 7.1 and merged its content into the main discussion section.

Reviewer 2 Report
Comments and Suggestions for Authors
The section provides a solid overview of background concepts, but it lacks a clear connection to the research objectives. A stronger framing of how this background directly supports your study would enhance its relevance.
Consider reducing redundancy between subsections to avoid overlap and improve focus.Technical terms, such as "FoA" (Focus of Attention), "RJA" (Responsive Joint Attention), and "JA" (Joint Attention), are used without consistent introductory explanations. Define each acronym clearly upon first use and avoid overloading the reader with jargon.
The introduction to the background is clear but could be more compelling. Explain why an interdisciplinary approach is critical to addressing gaps in Human-Robot Interaction (HRI).
You mention the Stereotype Content Model (SCM) but do not explain why this model was chosen over alternatives. Briefly justify its selection and relevance to HRI.
The discussion on gaze-following and imitation mechanisms is informative, but the practical challenges of implementing these mechanisms in real-world systems are underexplored. Include more critical insights or known limitations.
The connection between human social cognition and robot design is implicit. Explicitly state how findings from SCM research can be translated into robotic contexts.
The summary table (Table 1) is helpful but does not offer critical insights. Add commentary on recurring gaps or patterns in the studies reviewed, particularly concerning scalability or real-world applications of JA mechanisms.
Author Response
We have updated the manuscript content after carefully addressing all the reviewers' comments. We are uploading an updated manuscript where the changes have been highlighted using blue for the other reviewer and olive for you.
Comments 1: The section provides a solid overview of background concepts, but it lacks a clear connection to the research objectives. A stronger framing of how this background directly supports your study would enhance its relevance.
Response 1: We thank the reviewer for their insightful comment. While we intended to provide a thorough breakdown of the background concepts related to Human-Robot Interaction, joint attention, and user perception, we acknowledge that the connection to our research objectives was not made explicit enough. To address this, we have clarified how each subsection in the Background section directly contributes to our study.
Action 1: We have added clarifying statements in each subsection of Section 2 to explicitly link the background concepts to our research objectives. Below are the exact additions:
- Section 2.1 (User Perception of the Robot in HRI):
At the end of the section, "By integrating these social cognition metrics into our study, we aim to provide a deeper insight into how a social robot with RJA capabilities influences user perception across these dimensions while performing a task, affecting the quality of HRI." - Section 2.2 (Bio-Inspired Mechanisms of Joint Attention):
At the beginning of the section, "This section covers various bio-inspired mechanisms of joint attention and their implementations in robotics, serving as the foundation for designing our RJAS." and at the end of the section, “In this work, we propose integrating some of the previously described bio-inspired mechanisms --gaze-following, imitation, turn-taking, IOR, and vestibular movement-- in our RJAS to improve the robot's ability to engage in JA with humans. By leveraging these biological principles, the robot can prioritize stimuli, manage attention shifts, and maintain a natural interaction flow, providing a more intuitive and engaging HRI experience.” - Section 2.3 (Joint Attention in Human-Robot Interaction):
At the beginning of the section, "These gaps directly motivate our study, which aims to address this need through a bio-inspired system and rigorous evaluation using standardised metrics."
Comments 2: Consider reducing redundancy between subsections to avoid overlap and improve focus.
Response 2: We thank the reviewer for their constructive feedback. In response, we carefully reviewed the document and made adjustments to eliminate redundancies, with the most significant changes in Section 4.
Action 2: We have moved the subsection 4.2 to the end of the section and merged its content into the subsection 4.4. The resulting text is as follows:
“The main goal of the experimental setup is to create an immersive and realistic environment for HRI to evaluate the impact of RJA on the user's perception of the robot. For that, we decided to integrate the robot with the RJAS and distractive stimuli into an experimental scenario where the user plays with the robot in the ``Odds and Evens'' game. The ``Odds and Evens'' game is a classic chance-based game where two participants must choose either ``odds'' or ``evens''. Following a synchronised countdown, each player extends several fingers, and the sum of the fingers determines the outcome. If the total is an even number, the player who chose ``evens'' wins; if the total is odd, the ``odds'' player wins.
Figure 5 illustrates the interaction flow, starting with the robot presenting itself and explaining the game rules if needed. A calibration phase follows, where the robot ensures that the user is familiar with the display of numbers using their fingers. The game then proceeds with the user or robot selecting whether they will play as ``odd'' or ``even'' during the match and reminds the user of the basics of interacting with the robot to avoid false positives with the finger detection. Then, Mini guides the interaction, actively managing the countdown and informing the user of the game progress in real-time.
After the countdown, Mini displays its choice on the tablet, and the user has to display their hands. If the user does not show their fingers in time or the robot does not detect them accurately, the robot alerts the user and resets that round. Otherwise, the robot checks who the winner of this round is. This sequence repeats for five rounds until the match concludes with a winner. After each match, the robot checks if the user wishes to continue playing, concluding with a farewell when the game ends.
Auditory and visual distractive stimuli are introduced at different game stages to simulate a real-world scenario where the robot must respond to user distractions. These stimuli are carefully distributed throughout the interaction – reminder, countdown and check winner phases-- to avoid overstimulating the user and ensure that the primary focus remains on the game itself.”
Comments 3: Technical terms, such as "FoA" (Focus of Attention), "RJA" (Responsive Joint Attention), and "JA" (Joint Attention), are used without consistent introductory explanations. Define each acronym clearly upon first use and avoid overloading the reader with jargon.
Response 3: We agree that clearer definitions of these terms would enhance the paper's readability. In this sense, we acknowledge that their introduction was not detailed enough. To address this, we have reviewed these concepts in the manuscript and added more content upon the first use of each term.
Action 3: The following fragments have been added within each term in the Introduction (Section 1):
- Joint Attention (JA): “JA is a social-cognitive skill that enables individuals to align their focus with others, establishing a shared reference point. This alignment is often initiated or maintained through gaze-following, pointing gestures, or vocal cues. At its core, JA allows individuals to understand and respond to the attentional states of others, creating the foundation for collaboration, social learning, and communication.”
- Focus of Attention (FoA): “Here, the FoA refers to the specific stimulus or area within the environment that an individual prioritizes at any given time. For example, a sudden sound or motion may draw immediate attention, overriding less urgent stimuli.”
- Initiating Joint Attention (IJA): “IJA is an individual's ability to direct another's attention to a particular object or event. Communicative cues such as pointing, gesturing, or making verbal sounds often achieve this. For example, a person might point at a distant object and simultaneously look at their interaction partner to ensure they notice it.”
- Responding to Joint Attention (RJA): “Alternatively, RJA is the ability to recognise and follow the object or event referred by another person. This implies following the same cues mentioned for IJA. For instance, regarding the previous example, if one person looks at an object and/or gestures toward it, the observer aligns their focus with the indicated direction. Together, IJA and RJA form the building blocks of effective communication and collaborative interaction, enabling shared focus and mutual understanding.”
Comments 4: The introduction to the background is clear but could be more compelling. Explain why an interdisciplinary approach is critical to addressing gaps in Human-Robot Interaction (HRI).
Response 4: We appreciate the suggestion to strengthen the introduction to the background. In this regard, we have revised the text to provide a clearer justification for adopting this methodology, emphasizing how integrating insights from psychology, social cognition, and robotics allows us to address the complexities of Human-Robot Interaction (HRI) more effectively.
Action 4: The following fragment has been added to the beginning of Section 2: “To analyze the impact of RJA in HRI, we adopted an interdisciplinary approach, combining insights from various areas of psychology, social cognition, attention and robotics. This allows us to effectively address the human and robotic components, ensuring that the system aligns with natural human behaviours and cognitive processes.”
Comments 5: You mention the Stereotype Content Model (SCM) but do not explain why this model was chosen over alternatives. Briefly justify its selection and relevance to HRI. The connection between human social cognition and robot design is implicit. Explicitly state how findings from SCM research can be translated into robotic contexts.
Response 5: We thank the reviewer for their insightful comments regarding the SCM. In response, we have included a justification for the model selection, its relevance to HRI, and its connection to bridging human social cognition with robotic design. These aspects have been made explicit in the manuscript, ensuring the model's applicability and contributions are clearly articulated.
Action 5: We have added to Section 2 the following excerpt justifying SCM selection and how the findings from SCM research can be translated into robotic contexts: "The SCM was chosen for this study due to its robustness in evaluating social perception dimensions--competence and warmth--that are directly relevant to HRI. Competence reflects a robot's task efficiency and reliability, while warmth captures its friendliness and approachability. Findings from this framework can affect robotic design by emphasising how social cues, such as gaze direction or speech patterns, might influence user perceptions. For instance, a robot that maintains appropriate gaze contact could enhance its perceived warmth, while precise task execution may reinforce competence."
Comments 6: The discussion on gaze-following and imitation mechanisms is informative, but the practical challenges of implementing these mechanisms in real-world systems are underexplored. Include more critical insights or known limitations.
Response 6: We sincerely thank the reviewer for their thoughtful observation regarding the challenges of implementing gaze-following and imitation mechanisms in real-world systems. In response, we have expanded the discussion section to include additional technical and conceptual insights into the limitations and complexities associated with these mechanisms. These updates address both hardware constraints and broader philosophical considerations related to the perception and effectiveness of joint attention mechanisms.
Action 6: We have added the following fragments to the discussion (Section 7): “In addition to this, implementing JA mechanisms such as gaze-following and imitation in real-world systems presents technical and conceptual challenges. From a technical standpoint, achieving real-time operation requires the execution of an entire robotic architecture alongside the attention system. Our robot operates solely on CPU resources without internal GPU acceleration, thus requiring highly optimised vision models to sustain an acceptable frame rate during real-time interaction. Furthermore, vision-based head orientation estimations inherently involve some error due to the model itself, which propagates through the system and can affect the robot’s attentional responses.
On a conceptual level, replicating JA mechanisms in the robot does not guarantee that users perceive them as intended. Evaluating whether these mechanisms create the feeling of JA remains subjective and is challenging to measure. While questionnaires like RoSAS assess user perceptions of competence and warmth, tools like the Social Presence Inventory (SPI) could provide additional insights into whether users truly perceive shared focus from the robot.“
Comments 7: The summary table (Table 1) is helpful but does not offer critical insights. Add commentary on recurring gaps or patterns in the studies reviewed, particularly concerning scalability or real-world applications of JA mechanisms.
Response 7: We appreciate the reviewer's remarks regarding the need for critical insights related to the works in the summary table (Table 1). Indeed, many of the analysed works present recurring challenges, particularly in scalability and real-world application of JA mechanisms. We have added a detailed commentary in the manuscript addressing these gaps and showcasing how our work contributes to advancing the field.
Action 7: We have added a new paragraph at the end of Section 2, that addresses the recurring challenges identified in the reviewed studies as a summary. The included fragment is as follows: “These studies reveal challenges in implementing joint attention mechanisms in HRI. One of the main limitations is scalability, as many systems are tested in environments that do not account for the variability of real-world settings. In this sense, we aim to replicate a scenario where the user is exposed to stimuli that may arise during their interaction with the robot, encouraging them to shift their attention. This approach allows for a more effective evaluation of whether the attention mechanisms implemented in the robot are functioning as intended. Another observed pattern is the difficulty in ensuring robustness across different robotic platforms, as the hardware capabilities of robots significantly influence how the researcher is able to integrate JA mechanisms on the platform. Existing works often rely on custom or ad hoc robotic platforms explicitly designed for their experiments. This limits the general applicability of their findings to other systems.”

Round 2
Reviewer 2 Report
Comments and Suggestions for Authors
Accept